# Impact of Anti PD-1 Immunotherapy on HIV Reservoir and Anti-Viral Immune Responses in People Living with HIV and Cancer

**DOI:** 10.3390/cells11061015

**Published:** 2022-03-17

**Authors:** Marine Baron, Cathia Soulié, Armelle Lavolé, Lambert Assoumou, Baptiste Abbar, Baptiste Fouquet, Alice Rousseau, Marianne Veyri, Assia Samri, Alain Makinson, Sylvain Choquet, Julien Mazières, Solenn Brosseau, Brigitte Autran, Dominique Costagliola, Christine Katlama, Jacques Cadranel, Anne-Geneviève Marcelin, Olivier Lambotte, Jean-Philippe Spano, Amélie Guihot

**Affiliations:** 1INSERM U1135, CIMI, Département d’Immunologie, AP-HP, Hôpital Pitié-Salpêtrière, Sorbonne Université, F-75013 Paris, France; baptiste.abbar@aphp.fr (B.A.); baptiste.fouquet@aphp.fr (B.F.); alice.rousseau@upmc.fr (A.R.); assia.samri@upmc.fr (A.S.); brigitte.autran-ext@aphp.fr (B.A.); amelie.guihot@aphp.fr (A.G.); 2INSERM UMR_S 1136, Institut Pierre Louis d’Epidémiologie et de Santé Publique, Département de Virologie, AP-HP, Hôpital Pitié-Salpêtrière, Sorbonne Université, F-75013 Paris, France; cathia.soulie@inserm.fr (C.S.); anne-genevieve.marcelin@aphp.fr (A.-G.M.); 3GRC #04 Theranoscan, Département de Pneumologie et Oncologie Thoracique, AP-HP, Hôpital Tenon, Sorbonne Université, F-75020 Paris, France; armelle.lavole@aphp.fr (A.L.); jacques.cadranel@aphp.fr (J.C.); 4INSERM UMR_S 1136, Institut Pierre Louis d’Epidémiologie et de Santé Publique, Sorbonne Université, F-75013 Paris, France; lambert.assoumou@iplesp.upmc.fr (L.A.); dominique.costagliola@iplesp.upmc.fr (D.C.); 5Département d’Oncologie Médicale, AP-HP, Hôpital Pitié-Salpêtrière, Sorbonne Université, F-75013 Paris, France; marianne.veyri@aphp.fr (M.V.); jean-philippe.spano@aphp.fr (J.-P.S.); 6INSERM U1175, Département de Maladies Infectieuses, Centre Hospitalier Universitaire de Montpellier, Université de Montpellier, F-34090 Montpellier, France; a-makinson@chu-montpellier.fr; 7Département d’Hématologie Clinique, AP-HP, Hôpital Pitié-Salpêtrière, Sorbonne Université, F-75013 Paris, France; sylvain.choquet@aphp.fr; 8Département de Pneumologie, Centre Hospitalier Universitaire de Toulouse, F-31000 Toulouse, France; mazieres.j@chu-toulouse.fr; 9Département de Pneumologie, AP-HP, Hôpital Bichat-Claude Bernard, F-75018 Paris, France; solenn.brosseau@aphp.fr; 10Département de Maladies Infectieuses, AP-HP, Hôpital Pitié-Salpêtrière, Sorbonne Université, F-75013 Paris, France; christine.katlama@aphp.fr; 11Département d’Immunologie Clinique, AP-HP, Hôpital Bicêtre, Université Paris-Saclay, F-94270 Le Kremlin Bicêtre, France; olivier.lambotte@aphp.fr; 12INSERM, CEA, Center for Immunology of Viral, Auto-immune, Hematological and Bacterial Diseases (IDMIT/IMVA-HB), UMR1184, Université Paris-Saclay, F-94270 Le Kremlin Bicêtre, France

**Keywords:** immune checkpoint blockade, HIV reservoir, anti-HIV immune responses, compensatory mechanisms

## Abstract

The role of immune checkpoints (ICPs) in both anti-HIV T cell exhaustion and HIV reservoir persistence, has suggested that an HIV cure therapeutic strategy could involve ICP blockade. We studied the impact of anti-PD-1 therapy on HIV reservoirs and anti-viral immune responses in people living with HIV and treated for cancer. At several timepoints, we monitored CD4 cell counts, plasma HIV-RNA, cell associated (CA) HIV-DNA, EBV, CMV, HBV, HCV, and HHV-8 viral loads, activation markers, ICP expression and virus-specific T cells. Thirty-two patients were included, with median follow-up of 5 months. The CA HIV-DNA tended to decrease before cycle 2 (*p* = 0.049). Six patients exhibited a ≥0.5 log_10_ HIV-DNA decrease at least once. Among those, HIV-DNA became undetectable for 10 months in one patient. Overall, no significant increase in HIV-specific immunity was observed. In contrast, we detected an early increase in CTLA-4 + CD4+ T cells in all patients (*p* = 0.004) and a greater increase in CTLA-4+ and TIM-3 + CD8+ T cells in patients without HIV-DNA reduction compared to the others (*p ≤* 0.03). Our results suggest that ICP replacement compensatory mechanisms might limit the impact of anti-PD-1 monotherapy on HIV reservoirs, and pave the way for combination ICP blockade in HIV cure strategies.

## 1. Background

Immune checkpoints (ICPs), such as programmed cell death protein 1 (PD-1), cytotoxic T lymphocytes antigen 4 (CTLA-4), T-cell immunoreceptor with immunoglobulin and ITIM domain (TIGIT) and lymphocyte activation gene (LAG-3) are T cell inhibitory receptors that are activated on immune cells after TCR engagement to counterbalance chronic antigenic stimulation [1]. In cancers, ICPs are involved in immune control escape, promoting anti-tumor T-cell exhaustion. Nowadays, ICP blockade (ICB) by monoclonal antibodies is used for restoring anti-tumor immunity and is a major advance in cancer therapy [2].

During HIV-1 infection, ICPs are involved in both reservoir latency and anti-viral T cell exhaustion. On the one hand, PD-1, TIGIT and LAG-3 expression is positively associated with the frequency of CD4 + T cells harboring integrated HIV-DNA, and PD-1 engagement has been showed to inhibit HIV production and reactivation in latently infected cells ex vivo [3,4]. Accordingly, ICB reverses latency and reactivates viral production [3,5]. On the other hand, increased levels of PD-1 expression on total and HIV-specific CD8 + and CD4+ T cells in untreated HIV-1 infection are significantly correlated with both increased HIV viral load (VL) and with reduced capacity of cytokine production and of proliferation of HIV-specific T cells [6,7,8]. Anti-PD-1 and anti-PD-L1 antibodies demonstrated immune dysfunction reverse [8,9,10]. Taken together, these data suggest that ICB used as a “shock and kill” strategy could at the same time reverse latency and make the virus visible to the immune system, and restore anti-HIV immunity towards an HIV cure. The impact of ICB monotherapy on HIV-1 infection in vivo has been reported with controversies [11]. In our experience, one patient treated with nivolumab for cancer demonstrated a drastic and persistent decrease in cell-associated (CA) HIV-DNA [12]. Three other reports revealed latency reversal in patients [3,5,13]. A more recent study with very short follow-up including 33 people living with HIV (PLWH) with cancer showed no modification of the replication-competent reservoir [14].

During other co-infecting chronic viral infections, such as HBV, HCV, JC-virus or EBV infections, pathogens also exploit ICPs for immune evasion [15,16,17]. Several studies have suggested that ICB could be used as anti-infectious treatment for chronic HBV and HCV infection or for progressive multifocal leukoencephalopathy [18,19,20]. Finally, EBV-associated lymphomas and HHV-8- associated Kaposi sarcoma (KS) could benefit from ICB [21,22]. Of note, no predictive biomarker of ICB efficacy is available concerning chronic viral infections.

A biological sub-study was set-up, from both the ANRS CO24 OncoVIHAC prospective multicenter cohort (OncoVIRIM) and clinical trial IFCT-1602 CHIVA-2 (BIO-CHIVA-2), including PLWH with cancer treated with ICB. Here, we present the results of the largest cohort of patients assessing in vivo CA HIV-DNA, immunological changes and the evolution of virus-specific T cells among PLWH treated with ICB for cancer.

## 2. Methods

### 2.1. Study Design and Population

Patients were included from December 2017 to March 2020 from ten French centers on behalf the Agence nationale de recherche sur le SIDA et les hépatites virales (ANRS) and Intergroupe Francophone de cancérologie thoracique (IFCT) groups. Inclusion criteria: were age above 18 years old, proven HIV-1 or 2 infection, viral load below 50 cps/mL in OncoVIRIM or 200 cps/mL in BIO-CHIVA-2 under ART, ICB for cancer. Patients received nivolumab or cemiplimab 3 mg/kg every 2 weeks or pembrolizumab 2 mg/kg every 3 weeks until tumor progression, toxicity or patient decision to cease treatment. All patients signed a written informed consent form. The protocol was approved by an institutional review board (ANRS CO-24 OncoVIHAC: CPP17-020/2017-A00699-44; IFCT-1602 CHIVA2: CPP-Sud-Est V/17-IFCT-01) and was performed in accordance with the Helsinki declaration. Fresh blood samples were obtained at baseline and before cycles 2, 3 or 4, 9, 15 or 18, 27 or 36 and 51 and at the end of treatment.

### 2.2. HLA-Typing

HLA typing was performed using PCR-SSO hybridization methods in a Luminex FLEXMAP-3D^®^ platform and analyzed on the HLA-FusionTM software (LABtype SSO class I/II tests: LABtype XR Class I locus A, B, C and Labtype SSO class II DRB1, DQA1/B1, OneLambda Inc., West Hills, CA, USA).

### 2.3. Viral Assays

Serologies for CMV and EBV (DiaSorin, Saluggia, Italy), HBV and HCV (Abbott, Chicago, IL, USA), HHV-8 [23] were analyzed at baseline. The HIV, HBV and HCV viral loads (VL) were analyzed in plasma using AmpliPrep/COBAS TaqMan (Roche Diagnostics, Basel, Switzerland) and the CMV and EBV loads were measured in whole blood (Qiagen, Hilden, Germany). HHV-8 was quantified by RT-PCR [24]. CA HIV-1 DNA was quantified by ultrasensitive RT-PCR (Biocentric, Bandol, France) [25].

### 2.4. Flow Cytometric Analysis

Flow cytometry testing was performed on fresh blood with two mixes of antibodies assessing first lymphocyte subsets and ICP: TIM3-BB515 (BD, Franklin Lakes, NJ, USA, 7D3), CXCR5-PE (BioLegend, San Diego, CA, USA, J252D4), CD45RA-ECD (BC, Fullerton, CA, USA, 2H4LDH11LDB9), CD27-PerCP-Cy5.5 (BD, Franklin Lakes, NJ, USA, L128), CCR7-Pe-Cy7 (BD, Franklin Lakes, NJ, USA, 3D12), CTLA4-APC (BD, Franklin Lakes, NJ, USA, BNI3), CD4-APC-R700 (BD, RPA-T4, Franklin Lakes, NJ, USA), CD3-APC-Cy7 (BD, SK7, Franklin Lakes, NJ, USA), PD1-BV421 (BD, EH12.1, Franklin Lakes, NJ, USA) and CD8-BV510 (BD, SK1, Franklin Lakes, NJ, USA), and intracellular staining with CTLA4-APC (BD, BNI3, Franklin Lakes, NJ, USA); second, activation markers were assessed:CD4-FITC (BD, RPA-T4, Franklin Lakes, NJ, USA), CD25-PE (BC, B1.49.9, Fullerton, CA, USA), CD69-PE-CF594 (BD, FN50, Franklin Lakes, NJ, USA), PD1-PE-Cy7 (BD, EH12.1, Franklin Lakes, NJ, USA), HLA-DR-AF700 (BioLegend, San Diego, CA, USA, L243), CD3-APC-AF750 (BC, Fullerton, CA, USA, UCHT1), CD38-BV421 (BioLegend, San Diego, CA, USA, HB7) and CD8 BV510 (BD, Franklin Lakes, NJ, USA, SK1), and intracellular staining with Ki67-APC (ThermoFisher, Waltham, MA, USA, 20Raj1). Cell permeabilization (Cytofix/Cytoperm, BD, Franklin Lakes, NJ, USA), fixation (CellFIX, BD, Franklin Lakes, NJ, USA), acquisition (10-colors Gallios) and analysis (FlowJo 10.5) were performed as previously described [26]. T cell subsets were defined as: naive T cells (TN): CD27 + 45RA + CCR7+, central memory T cells (TCM): CD27 + 45RA-CCR7+, transitional memory T cells (TTM): CD27 + 45RA-CCR7-, effector memory T cells (TEM): CD27-45RA-CCR7-, CD45RA re-expressing TEM (TEMRA): CD27-45RA + CCR7-. Boolean ICP analysis defined single, double and triple expressers displaying only one, two or three ICP.

### 2.5. Intracellular Cytokine Staining Assays

Virus-specific T cells were analyzed using an ICS assay [27]. Briefly, 1 × 10^6^ thawed PBMC were stimulated overnight with each virus peptides pool (Appendix A) or staphylococcal enterotoxin B toxin as the positive control and medium alone as negative control. Cells were surface-stained with Live/Dead Fixable Aqua Dead Cell (Invitrogen, Carlsbad, CA, USA, L34957) and anti PD1-BV421 (BD, Franklin Lakes, NJ, USA, EH12.1), CD4-APC (BD, Franklin Lakes, NJ, USA, RPA-T4) and CD8-PerCP-Cy5.5 (BD, Franklin Lakes, NJ, USA, SK1); then intracellular staining was performed for interleukin-2 (IL2)-PE (BD, Franklin Lakes, NJ, USA, 5344.111), interferon-γ (IFN-γ)-FITC (BD, Franklin Lakes, NJ, USA, 25723.11), tumor necrosis factor-α (TNF-α)-PE-Cy7 (BD, Franklin Lakes, NJ, USA, Mab11) and CD3-APC-Cy7 (BD, Franklin Lakes, NJ, USA, SK7).

The HIV-stimulated cells from 18 patients were also analyzed for the expression of ICP and surface stained with anti CD4-ECD (BC, Fullerton, CA, USA, UCHT1, 7448079F), CD8-PerCP-Cy5.5 (BD, Franklin Lakes, NJ, USA, SK1), PD1-BV421 (BD, Franklin Lakes, NJ, USA, EH12.1), TIM3-PeCy7 (ThermoFisher; Waltham, MA, USA, F38-2E2), CTLA4-APC (BD, Franklin Lakes, NJ, USA, BNI3), LAG3-AlexaFluor700 (ThermoFisher, Waltham, MA, USA, 3DS223H) and IgG 4-PE (Southern Biotech, Birmingham, AL, USA, HP6025); then intracellular staining was performed for IFN-γ-FITC (BD, Franklin Lakes, NJ, USA, 25723.11), CD3-APC-Cy7 (BD, Franklin Lakes, NJ, USA, SK7) and CTLA4-APC (BD, Franklin Lakes, NJ, USA, BNI3). 

Percentages of CD8+ and CD4 + T cells producing cytokines were determined after subtraction of negative controls and by adding the various peptide pools for each virus. Phenotypes of specific T cells were evaluated only on cytokine-producing cells with more than 50 events. 

### 2.6. Statistical Analysis

The non-parametric Wilcoxon test was used to test differences between paired groups. The Mann–Whitney test was used to test differences between unpaired groups. Statistical significance was considered for *p*-values below 0.05. When multiple comparisons were performed, the Bonferroni correction was used to correct the significance level, as stated in the figure legends.

## 3. Results

### 3.1. Patients Characteristics

Thirty-two patients with HIV-1 infection were included: *n* = 22 in OncoVIRIM, *n* = 10 in BIO-CHIVA-2 (Table 1). The cancer types were non-small cell lung-cancer (*n* = 20), bladder cancer (*n* = 3), melanoma (*n* = 2), head and neck cancer (*n* = 2), Hodgkin lymphoma, KS, anal, oropharynx and orbit cancer (1 patient each). Viral co-infections included HHV-8 infection (53%), resolved HBV infection (38%), resolved HCV infection (38%) and chronic HBV infection (22%). At baseline, the median CA HIV-DNA was 184 cps/10^6^ cells (range 40–1749) and the median HIV VL was 20 cps/mL (range <1–352). The median baseline CD4 cell count was 369/mm^3^ (range 45–915) and the CD4/CD8 ratio was 1 (range 0.2–2.1). All patients were treated with anti-PD-1: nivolumab (69%), pembrolizumab (28%) and cemiplimab (3%). Six of the patients received ICB as a first line treatment and the others had refractory or relapse diseases. The median follow-up duration was five months (range 1–30) and the median number of cycles received was six (range 2–36). At last follow-up, all but one patient had discontinued ICB. Eighteen patients died: 15 from tumor progression, one from an immune-related adverse event (myocarditis), one from COVID-19 infection and one from an unknown cause.

### 3.2. Early Two-Fold Decrease of CA-HIV-DNA following Anti-PD-1

Evaluating as a first step the total CA HIV-DNA at each time point, we observed a 1.9-fold decrease from a median 184 to 99 cps/10^6^ cells at C2 when compared to baseline (*p* = 0.0499), without reaching the significance level after the Bonferroni correction (Figure 1). In addition, as 0.3 log_10_ physiological fluctuations are commonly described in this assay, we used a threshold of 0.5 log_10_ to further evaluate this decay. A more than 0.5 log_10_ decrease in HIV-DNA cps/10^6^ cells was observed at once after anti-PD-1 initiation in six patients (Pt #7, #14, #17, #21, #30 and #31). We further identified these patients with reservoir size reduction as “RR” patients, and we studied if they exhibited specific immunological characteristics to better understand which factors could be involved. Baseline CA HIV DNA was higher in RR compared to those without reservoir size reduction (further identified as “NoRR” patients): 307 versus 166 cps/10^6^ cells (*p* = 0.002) (Appendix A). Amongst the six RR, patient #7 demonstrated a persistent and stable HIV-DNA decrease below the detection threshold at the three consecutive last samples. Among the 26 other patients, 4 had a transient increase of ≥0.5 log_10_ HIV-DNA cps/10^6^ cells and there was no change below or above 0.5 log_10_ for the 22 other patients. Because age potentially impacts the expression of PD-1 on T lymphocytes, we looked at baseline HIV-DNA between younger (<60 years) and older (≥60 years) patients and found no difference between groups (median values 191 and 181 cps/10^6^ cells, respectively, *p* = 0.9018). We also compared the median age in RR and in NR and found no significant difference (*p* = 0.6974).

### 3.3. CD4 Stability and Early T Cell Activation

There was no difference in baseline CD4 count and HIV-VL between RR and NoRR patients (Appendix A). The HIV-VL decreased from a median baseline of 20 cp/mL to 1 cp/mL at cycle 9 (*p* = 0.0313) without reaching the significance level after the Bonferroni correction, and there was no change in the CD4 cell counts (Figure 1), nor in the CD4/CD8 ratios, in the CD3, CD8, NK and B cell counts or in lymphocyte differentiation (Appendix A). Overall, the proportions of HLA-DR + CD4+ and CD8 + T cells and of CD38+ and HLA-DR + CD38 + CD8 + T cells significantly increased at C2 or C3: respectively 1.8, 1.6, 1.4 and 1.7 fold increase from baseline (*p* < 0.0001, Figure 2, Appendix A). Similarly, there was an early increase in Ki67 + CD4+ and CD8 + T cells at C2 or C3: respectively 2.1 and 2.3 fold increase from baseline (*p* ≤ 0.0003). Among RR, the baseline proportions of CD25 + CD8+ T cells (33%) in patient #7 and of HLA-DR + CD8 + T cells (25%) in patient #21 were the highest of the cohort, while patient #14 displayed an early major increase of CD4 + Ki67+ and CD4 + HLA-DR T cells at C3. Nevertheless, there was no statistical difference between activation markers at baseline or kinetics among patients with and without reservoir size reduction (Appendix A). These results confirm that ICB induces early immunological activation and show that some RR patients had high baseline levels of activation markers.

### 3.4. CTLA-4 Is Upregulated on CD4 Cells and ICP Compensatory Mechanisms Are Less Pronounced in Patients RR

Next, we assessed at each time point the co-expression of PD-1 together with CTLA-4 and TIM-3 on T cells to explore whether inhibitory compensation of ICP blockade occurred (Figure 3, Appendix A). At baseline, RR patients did not demonstrate any specific profile except patient RR #21 who showed the highest proportion of PD-1 + CD8 + T cells (99%) in the cohort. (Appendix A). At C2 and overtime, the membrane PD-1 molecule became barely detectable on CD4+ and CD8 + T cells (*p* < 0.0001). In contrast, we observed a significant but transient increase in CTLA-4 + CD4 + T cells at C2 (fold change: 1.5, *p* = 0.004) and a trend for an increase of CTLA-4 + CD8+ T cells at C2 which did not reach statistical significance (*p* = 0.03). Of note, the proportions of CD8+ T cells displaying CTLA-4 and TIM-3 increased more at C2 or C3 in NoRR patients compared to RR patients (respectively 2.4 versus 1.1 for CTLA-4 + CD8 + fold change, and 1.4 versus 0.9 for TIM-3 + CD8 + fold change) (*p* ≤ 0.03) (Figure 3). Taken together, these results suggest that anti-PD-1 monotherapy could be associated with some compensatory increase of other ICP molecules on T cells which is less pronounced in the RR patients.

### 3.5. Stability of Peripheral HIV-Specific T Cells Despite PD-1 Overexpression at Baseline

We next evaluated whether ICB could enhance HIV-specific T cell responses in vivo, defined as IFN-γ producing CD8+ T cells. At baseline, the median proportion of HIV-specific CD8 + T cells was 0.48% (range 0–6.52) of total CD8+ T cells. Those frequencies did not differ in the RR patients compared to the NoRR patients, although the patient RR #21 had the third highest proportion of HIV-specific CD8+ T cells among the cohort (2.63%) (Appendix A). Overall, the percentages of HIV-specific CD8+ T cells did not statistically change over time (Figure 4) and there was no difference in the kinetics of HIV-specific CD8+ T cells between the RR and NoRR patients (Appendix A). The frequencies of IL-2 + CD8+ and TNF-α+ CD8+ T cells were much lower at baseline (0.02% each) than the IFN-γ + CD8+ T cells frequencies and did not change over time (Appendix A). The cell poly-functionality defined as the co-production of IFN-γ, and/or IL-2 and/or TNF-α also remained stable over time without differences between RR patients and the others (Appendix A). Similarly, CD4 HIV-specific T cell responses did not change over time (Appendix A). 

When analysing PD1 expression on HIV-specific T cells at baseline, we observed that HIV-specific T cells displayed higher percentages of PD-1+ cells and CTLA-4+ cells (Figure 5 and Appendix A) and higher mean fluorescence intensity (MFI) of PD-1 (Appendix A) compared to non-HIV-specific CD8+ T cells (*p* < 0.001). PD-1 expression on HIV-specific CD8 + T cells was non significantly lower in RR compared to NoRR patients (Appendix A). We then tested in 18 patients whether such a lack of immune enhancement could reflect an upregulation of other ICP on HIV-specific CD8 T cells. However, CTLA-4, TIM-3 and LAG-3 expression on HIV-specific T cells did not increase over time (Figure 5). 

Taken together, these results suggest that ICB monotherapy is insufficient for enhancing an HIV-specific response despite high PD1-expression on HIV-specific CD8+ T cells at baseline. 

### 3.6. Immunological and Virological Parameters of Other Viruses Are Not Modified

As ICB have been proposed to be used against other chronic viral co-infections such as EBV, CMV and HBV, we aimed at evaluating the effects of ICB on other viruses in these poly-infected patients, and whether anti-PD-1 could enhance these other virus-specific T cell responses. In patients seropositive for HHV-8, HCV, HBV and CMV, there was no corresponding viral replication at baseline, except in 2 patients with active HBV infection and for whom anti-HBV treatment was changed at the start of ICB. After anti-PD-1 initiation, there was no significant change in HHV-8, HBV, HCV, EBV and CMV loads over time (Appendix A). There was no significant change in virus-specific CD8 + T cells over time despite highly heterogeneous profiles (Figure 6). When studying PD-1 expression on virus-specific CD8+ T cells, we observed at baseline higher percentages of PD-1 expression (Figure 5) and higher MFI of PD-1 (Appendix A) on EBV- and HBV-specific CD8+ T cells (*p* < 0.007). Taken together, these results suggest that ICB can be safely used in PLWH with viral co-infections and that PD-1 overexpression on EBV and HBV-specific CD8+ T cells could be targeted, even though we failed to demonstrate specific invigoration in peripheral blood.

### 3.7. Immuno-Virological Profiling of RRs

The individual profiles (HIV-RNA, CA HIV-DNA and HIV-specific T cells) of RR patients are summarized in Figure 7. None of these patients with available HLA typing (*n* = 4) had the HLA-B27 or B57 protective alleles. Patient #7 had the longest follow-up of 14 months and his HIV-DNA decreased from 187 cps/10^6^ cells to become persistently undetectable at the last three time points (C9, 15 and C27). Patient #14 displayed the highest HIV-DNA at baseline (1749 cps/10^6^ cells) which decreased by 0.6 log_10_ down to 403 cps/10^6^ cells six months later. Patient #17 experienced an early 0.5 log_10_ decrease in HIV-DNA from 620 to 178 cps/10^6^ cells at C2. Patient #21 demonstrated a persistent CA HIV-DNA reduction from 231 to 66 cps/10^6^ cells 4 months later (−0.5 log_10_). Patient #30 HIV-DNA decreased from 363 to 106 cps/10^6^ cells (−0.5 log_10_) 4 months later. Finally, the HIV-DNA of patient #31 decreased gradually from 251 to 53 cps/10^6^ cells 2 months later (−0.7 log_10_). ICB was stopped because of tumor progression in 4 cases and of cutaneous toxicity in 1 case, and monitoring was stopped in 1 case because of patient relocation.

## 4. Discussion

Here, we report the largest comprehensive and homogeneous immunological and virological assessment of the influence of immune check-point blockade on HIV reservoirs and T cell functions in a series of 32 PLWH treated with anti-PD-1 for cancer. 

Despite some cancers heterogeneity, there was a majority of lung cancers and no evidence that the cancer type might influence the ICB action on the HIV reservoir and on HIV-specific T cells. 

We chose to focus as a first step of reservoir size analysis on the measurement of total CA HIV-DNA since our final aim was to study the evolution of the HIV-reservoir as a result of both HIV latent cell reactivation and immune invigoration. We assume that it provides less understanding of the potential influence of ICB on the reservoirs, but such evaluation appeared to us to be more relevant on a clinical perspective. The CA HIV-DNA decrease observed after the cycle 1 was only transient and such ICB could have impacted more profoundly and durably the HIV-reservoir, though still transiently, in only 19% of cases. The threshold of CA HIV-DNA reduction of 0.5 log is above the usual 0.3 log_10_ fluctuations of the methods and was an acceptable cut-off in other studies [28,29]. This proportion of 19% RR could be mis-estimated given that CA HIV-DNA was undetectable at baseline in six patients. The 6 patients with a CA HIV-DNA decrease ≥ 0.5 log_10_ under ICB included one with CA HIV-DNA becoming persistently undetectable for 14 months and one with features of a “shock and kill”. Although follow-up was short due to frequent tumor progression, four patients with RR were followed during more than three months, which represents one of the longest descriptions of CA-HIV DNA under ICB. Of note, two out of six patients RR (patient #14 and #17) had a decrease of CD4+ T cells frequencies at the same time, which may have impacted the results of HIV reservoir. Overall, no significant immunological differences in terms of immune activation or HIV-specific T cell responses were observed between these rare RR and NoRR patients. The weak HIV-specific T cell responses observed in our patients and the lack of ICB effect might reflect the consequences of previous multiple chemotherapies in those PLWH with cancers 

Our results suggest one mechanism to explain such limited ICB impact on HIV reservoirs is that the virus persists preferentially in CD4 + T cells expressing other ICP than PD-1. The greater increase in CTLA-4 and TIM-3 expression on CD8 + T cells in the NoRR compared to the RR patients reinforces this hypothesis. Such ICP compensation and upregulation on CD8 + T cells could have hampered the CTL response to the virus and prevent an efficient “shock and kill” mechanism. These observations are consistent with the recently published demonstration of ICP overexpression in Hodgkin lymphoma tumor microenvironment exposed to anti-PD-1 [30]. Thus, a combination of IC blockade could be necessary to bypass multiple ICP expression both on HIV reservoir cells and on HIV-specific T cells as suggested by others, although such combinations are limited by an increased and barely acceptable toxicity in PLWH [4,14,31,32,33]. Finally, the dramatic decrease in PD-1 expression on T cells reflects the antigenic occupancy with the monoclonal antibody of detection which recognizes the same epitope as the therapeutic one, as it has been demonstrated previously [34]. 

Finally, we provide insights into the virological safety of ICB and its impact on immune responses against other viruses. Indeed, ICB is safe and can be widely used in co-infected patients, although the great majority of patients had very low viral antigen loads with undetectable HBV, HCV, CMV and HHV-8 VL at baseline. Of note, the two patients with active HBV infection at baseline benefited from anti-HBV treatment at the same time, which also hinders the interpretation of VL decrease under ICB.

In conclusion, our study shows that ICB with anti-PD1 antibodies in PLWH with cancer had a very limited impact on HIV reservoirs and immunity to HIV which might be explained by an ICP compensatory phenomenon as assessed by the early increase in CTLA-4 and Tim-3 expression. Further studies are warranted to evaluate the impact of combined immune checkpoint blockade on the HIV reservoirs and immunity of PLWH suffering from cancers.

## Figures and Tables

**Figure 1 cells-11-01015-f001:**
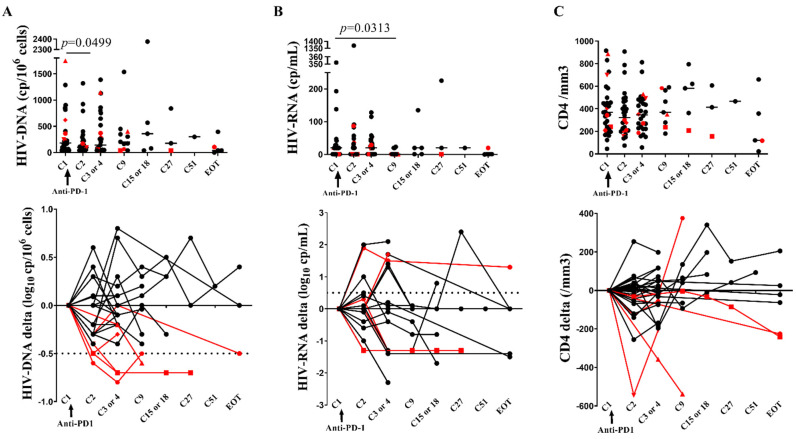
Anti-PD-1 therapy in PLWH with cancer does not significantly impact the biology of HIV. (**A**), HIV-DNA (cp/10^6^cells) (upper panel) and HIV-DNA (log_10_ cp/10^6^ cells) delta from baseline (lower panel), (**B**), HIV-RNA (cp/mL) (upper panel) and HIV-RNA (log_10_ cp/mL) delta from baseline (lower panel), (**C**), CD4 count (/mm^3^) (upper panel) and CD4 count (/mm^3^) delta from baseline (lower panel). Dotted lines denote decrease ≤ 0.5 log_10_ for CA HIV-DNA and increase ≥ 0.5 log_10_ for HIV-RNA. Red color denotes the RR patients represented with different shapes. EOT: end of treatment. Bonferroni corrected *p*-value: *p* ≤ 0.007, Wilcoxon-matched pairs signed Rank test. At cycle 1, *n* = 32; at cycle 2, *n* = 29; at cycle 3 or 4, *n* = 29; at cycle 9, *n* = 11; at cycle 15, *n* = 5; at cycle 27, *n* = 3; at cycle 51, *n* = 1; at end of treatment, *n* = 5. Experiments were monoplicates.

**Figure 2 cells-11-01015-f002:**
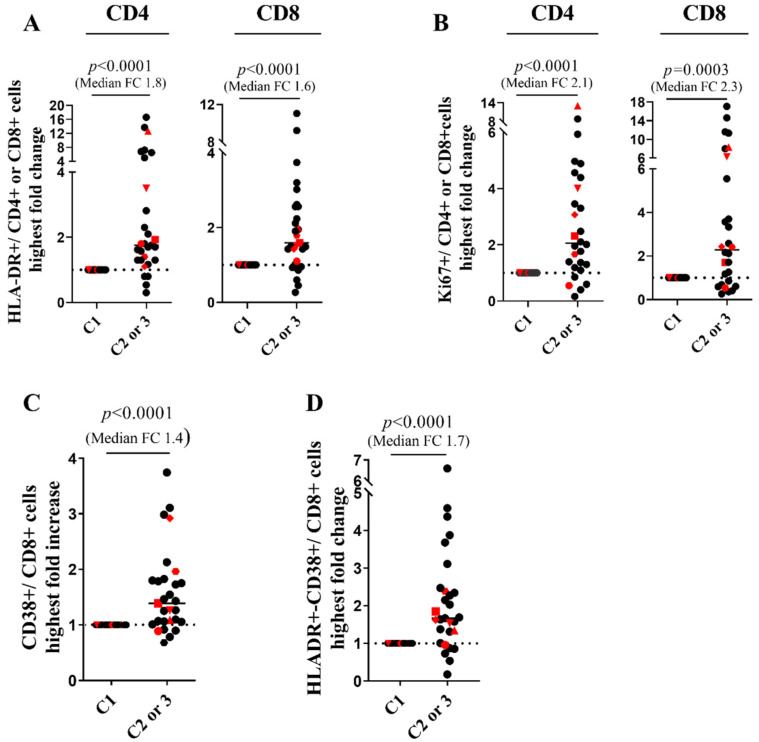
Anti-PD-1 treatment in PLWH with cancer results in T cell immunological activation. (**A**), Highest fold change at C2 or 3 from baseline in HLA-DR + cells among CD4+ and CD8 + T cells. (**B**), Highest fold change at C2 or 3 from baseline in Ki67+ cells among CD4+ and CD8+ T cells. (**C**), Highest fold change at C2 or 3 from baseline in CD38+ cells among CD8+ T cells. (**D**), Highest fold change at C2 or 3 from baseline in CD38 + HLA-DR + cells among CD8 + T cells. Red spots denote the RR patients and dotted lines denote fold change of 1. Wilcoxon- signed Rank test. FC: fold change. Experiments were monoplicates.

**Figure 3 cells-11-01015-f003:**
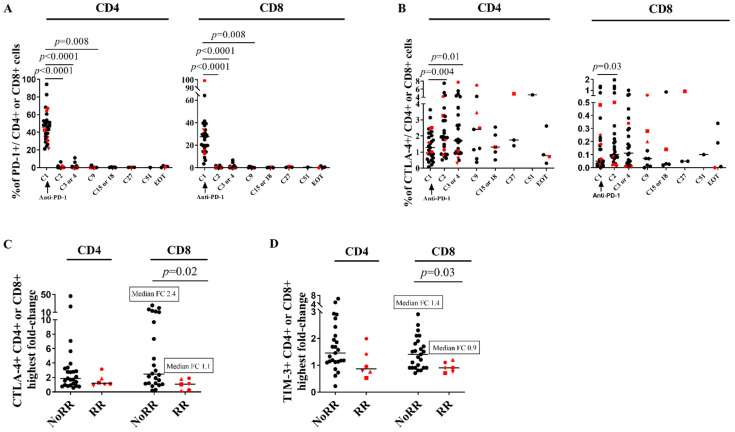
Anti-PD-1 therapy in PLWH with cancer induces dramatic decrease of PD-1 detection and slight and transient CTLA-4 expression increase. (**A**), Changes overtime in PD-1 expression on CD4 + (left) and CD8 + (right) T cells. (**B**), Changes overtime in CTLA-4 expression on CD4 + (left) and CD8 + (right) T cells. Bonferroni corrected *p*-value: *p* ≤ 0.007, Wilcoxon-matched pairs signed Rank test. Highest fold change at cycle 2 or 3 of (**C**), CTLA-4 + CD4 + (left) and CD8 + (right) T cells, and of (**D**), TIM-3 + CD4 + T (left) and CD8 + (right) among the patients without reservoir reduction (NoRR) and with reservoir reduction (RR). Mann–Whitney test. In each graph, red spots denote the patients RR. EOT: end of treatment. FC: fold change. At cycle 1, *n* = 32; at cycle 2, *n* =29; at cycle 3 or 4, *n*= 29; at cycle 9, *n* = 11; at cycle 15, *n* =5; at cycle 27, *n* = 3; at cycle 51, *n* = 1; at end of treatment, *n* = 5. Experiments were monoplicates.

**Figure 4 cells-11-01015-f004:**
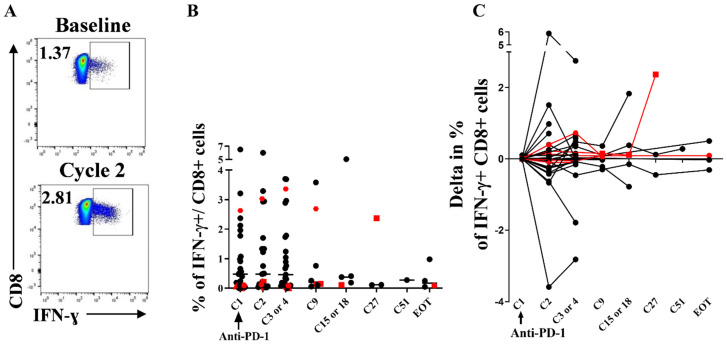
Anti-PD-1 therapy in PLWH with cancer does not significantly increase the in vivo frequency of HIV-specific CD8 + T cells. Thawed PBMC were stimulated during 6 h with HIV peptides (15 mers overlapping peptides covering RT, Nef and Gag) and then stained with intra-cellular anti-cytokine antibodies (anti-IFN-ɣ, anti- IL-2 and anti-TNF-α). HIV-specific CD8 + T cells were defined as IFN- ɣ producing CD8 + T cells after HIV stimulation. (**A**), Example dot plots of IFN-ɣ expression among CD8 + T cells after HIV stimulation at baseline and at cycle 2. (**B**), frequency of HIV-specific CD8+ T cells under ICB. (**C**), HIV-specific CD8 + T cells delta from baseline under ICB. Red colors denote the RR patients. EOT: end of treatment. At cycle 1, *n* = 32; at cycle 2, *n* =29; at cycle 3 or 4, *n*= 29; at cycle 9, *n* = 11; at cycle 15, *n* =5; at cycle 27, *n* = 3; at cycle 51, *n* = 1; at end of treatment, *n* = 5. Experiments were monoplicates.

**Figure 5 cells-11-01015-f005:**
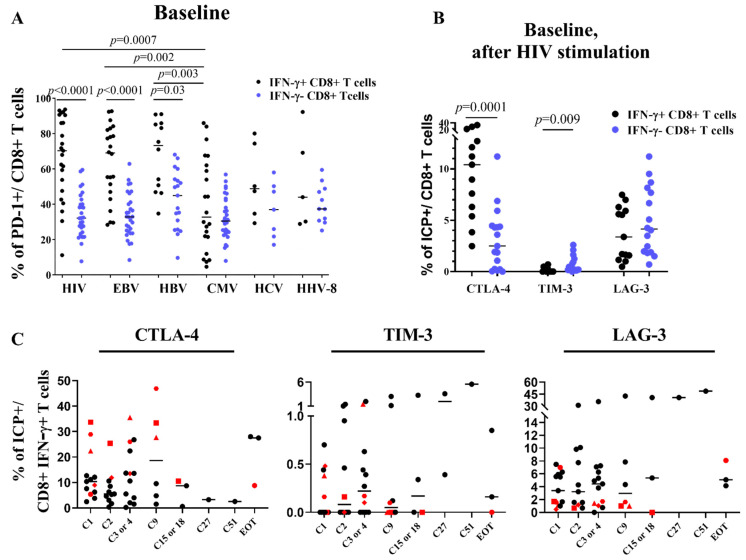
In PLWH with cancer, high levels of ICP expression are detected on HIV-specific T cells, without upregulation on anti-PD-1 therapy. Thawed PBMC were stimulated during 6 h with viral peptides and then stained with intra-cellular anti-cytokine antibodies (anti-IFN-ɣ, anti-IL-2 and anti-TNF-α). HIV-specific CD8 + T cells were defined as IFN- ɣ producing CD8+ T cells after HIV stimulation. (**A**), PD-1 expression among specific (black spots) and non-specific (blue square) CD8 + T cells at baseline depending on viral stimulations. Mann Whitney test. (**B**), CTLA-4, TIM-3 and LAG-3 expression among HIV-specific and non-specific CD8 + T cells after HIV stimulation (*n* = 18). Mann Whitney test. (**C**), expression of CTLA-4 (left), TIM-3 (middle) and LAG-3 (right) on HIV specific CD8 + T cells overtime. Red spots denote the patients with reservoir reduction. EOT: end of treatment. At cycle 1, *n* = 32; at cycle 2, *n* =29; at cycle 3 or 4, *n*= 29; at cycle 9, *n* = 11; at cycle 15, *n* =5; at cycle 27, *n* = 3; at cycle 51, *n* = 1; at end of treatment, *n* = 5. Experiments were monoplicates.

**Figure 6 cells-11-01015-f006:**
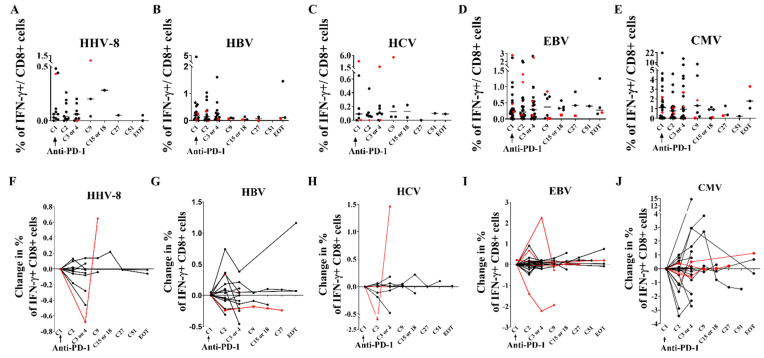
Anti-PD-1 therapy in PLWH with cancer does not significantly increase the frequency of virus-specific CD8 + cells. Thawed PBMC were stimulated during 6 h with viral peptides and then stained with intra-cellular anti-cytokine antibodies (anti-IFN-ɣ, anti-IL-2 and anti-TNF-α). Virus-specific CD8 + T cells were defined as IFN- ɣ producing CD8+ T cells after viral stimulation. Frequencies of (**A**), HHV-8, (**B**), HBV, (**C**), HCV, (**D**), EBV, and (**E**), CMV-specific-CD8+ T cells overtime. Delta from baseline of (**F**), HHV-8, (**G**), HBV, (**H**), HCV, (**I**), EBV, and (**J**), CMV-specific CD8 + T cells under ICB. Red colors denote the patients with reservoir reduction. EOT: end of treatment. Experiments were monoplicates.

**Figure 7 cells-11-01015-f007:**
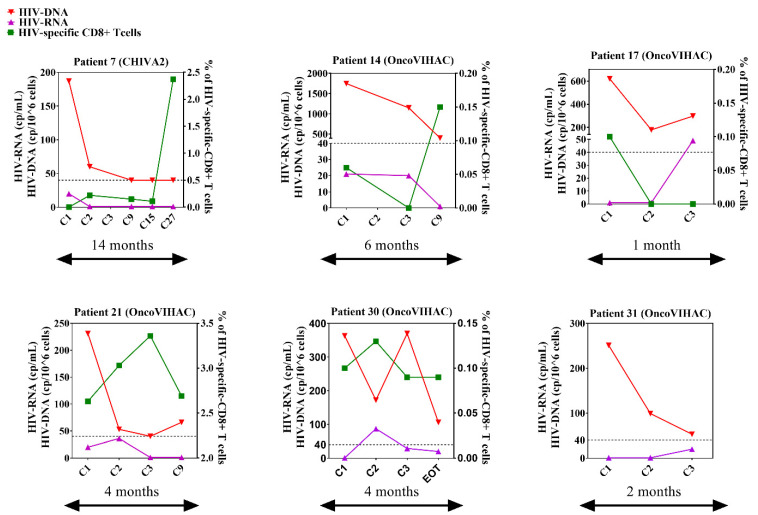
Immunovirological- profiles of the six PLWH patients treated with anti-PD-1 with a CA HIV-DNA decrease ≥ 0.5 log_10_. In each graph, red inverted triangles represent HIV-DNA (cp/10^6^ cells), purple triangles represent HIV-RNA (cp/mL) and green squares represent HIV-specific CD8 + cells (% of total CD8 + T cells). Dotted lines represent CA HIV-DNA = 40 cp/10^6^ cells, which is the detection threshold. EOT: end of treatment. The frequency of HIV-specific CD8 + T cells was not available for patient #31. Experiments were monoplicates.

**Table 1 cells-11-01015-t001:** Clinical and biological characteristics at baseline.

Pt	Cohort	Age	Sex	Type of Cancers	Pv Line	ICB Type	ART	CD4 Count (/mm^3^)	CD4/CD8 Ratio	HIV VL (cp/mL)	HIV-DNA(cp/10 cells)	HLA Typing
1	CHIVA-2	71	M	NSCLC	2	Nivo	Rilpivirine, Dolutegravir	183	0.5	28	<40	A*01:03 B*37:49
2	CHIVA-2	68	M	NSCLC	2	Nivo	Abacavir, Nevirapine	NA	NA	<1	227	A*03:11 B*07:27
3	CHIVA-2	56	M	NSCLC	1	Nivo	Emtricitabine tenofovir disoproxil, Maraviroc	831	0.4	193	818	A*24:29 B*38:44
4	CHIVA-2	59	M	NSCLC	1	Nivo	Emtricitabine, Rilpivirine, Tenofovir alafenamide	596	0.9	<1	<40	A*02:29 B*49:58
5	CHIVA-2	65	M	NSCLC	2	Nivo	Tenofovir, Emtricitabine, Bictegravir	451	0.9 *	352	904	A*02:29 B*07:15
6	CHIVA-2	55	F	NSCLC	1	Nivo	Dolutegravir, Abacavir, Lamivudine	499	0.6	138	1287	A*29:30 B*37:44
7	CHIVA-2	68	M	NSCLC	1	Nivo	Abacavir, Efavirenz	241	0.7	<20	187	A*02:11 B*39:40
8	CHIVA-2	53	F	NSCLC	1	Nivo	Efavirenz, Emtricitabine, Tenofovir	291	0.4	<20	66	A*02:66 B*49:52
9	CHIVA-2	58	M	NSCLC	1	Nivo	Dolutegravir, Abacavir, Lamivudine	249	0.2	<20	851	A*01:03 B*08:51
10	CHIVA-2	59	M	NSCLC	1	Nivo	Lamivudine, Dolutegravir, Abacavir	583	0.4	<1	231	A*02:03 B*18 27
11	Onco VIHAC	62	M	Melanoma	0	Nivo	Elvitegravir, Emtricitabine, Tenofovir	455	0.8	<1	166	A*02:26 B* 07:08
12	Onco VIHAC	69	M	NSCLC	0	Pembro	Emtricitabine, Rilpivirine, Tenofovir	273	0.4	<1	<40	A*02:24 B*35:57
13	Onco VIHAC	75	M	NSCLC	1	Nivo	Lamivudine, Dolutegravir	217	1	42	218	A*25:31 B*40:51
14	Onco VIHAC	60	F	NSCLC	0	Pembro	Darunavir, Norvir, Raltegravir	888	2.1	21	1749	A*31:68 B*07:07
15	Onco VIHAC	63	M	NSCLC	1	Nivo	Dolutegravir, Abacavir, Lamivudine	238	1.8	<1	<40	A*03:23 B*44:53
16	Onco VIHAC	53	M	HL	3	Nivo	Dolutegravir, Lamivudine	373	0.3	<20	173	A*33:68 B*14:44
17	Onco VIHAC	53	M	NSCLC	2	Nivo	Abacavir, Lamivudine	405	0.6	<1	620	A*02:29 B*40:49
18	Onco VIHAC	64	M	Bladder	2	Pembro	Abacavir, Lamivudine, Nivérapine	449	1.1	<1	213	A*02:31 B*07:40
19	Onco VIHAC	62	M	Oropharynx	2	Nivo	Darunavir, Ritonavir	162	0.5	<20	80	A*02:11 B*15:40
20	Onco VIHAC	58	M	Kaposi Sarcoma	4	Nivo	Dolutegravir, Abacavir, Lamivudine	728	2.1	47	409	A*33:68 B*14:44
21	Onco VIHAC	62	M	Anal	2	Nivo	Dolutegravir, Lamivudine	209	1.2	<20	231	A*02:24 B*35:44
22	Onco VIHAC	52	M	Head and neck	1	Nivo	Darunavir, Norvir, Raltegravir	369	0.8	<20	191	A*30:33 B*07:15
23	Onco VIHAC	71	F	Head and neck	1	Nivo	Dolutegravir	333	1.1	<1	<40	A*02:24 B*44:50
24	Onco VIHAC	63	M	Eye	2	Cemi	Bictegravir, Emtricitabne, Tenofovir	45	0.2	29	<40	A*02:33 B*14:53
25	Onco VIHAC	70	M	Melanoma	0	Pembro	Emtricitabine, Tenofovir, Névirapine	434	0.5	<1	181	A*01:03 B*07:51
26	Onco VIHAC	56	M	NSCLC	2	Pembro	Emtricitabine, Tenofovir, Darunavir, Ritonavir	424	0.8	<20	166	A*11:11 B*15:27
27	Onco VIHAC	62	M	Bladder	1	Pembro	Efavirenz, Emtricitabine, Tenofovir	915	1,4	<20	99	A*24:24 B*44:44
28	Onco VIHAC	58	M	NSCLC	0	Pembro	Darunavir, Doletugravir, Ritonavir, Tenofovir	192	0,4	<1	73	A*02:32 B*44:51
29	Onco VIHAC	62	M	NSCLC	1	Nivo	Raltegravir, Emtricitabine, Tenofovir	534	0.3	<1	99	NA
30	Onco VIHAC	60	M	Bladder	2	Pembro	Bictegravir, Emtricitabne, Tenofovir	969	0.6	<1	363	NA
31	Onco VIHAC	59	M	NSCLC	1	Nivo	Darunavir, Ritonavir	699	1	<1	251	NA
32	Onco VIHAC	60	M	NSCLC	0	Pembro	Bictegravir, Emtricitabne, Tenofovir	169	0.4	<1	59	NA
all	OncoVIHAC 69%	61	M 88%	NSCLC 63% Bladder 9%	1	Nivo 69% Pembro 28%		369	1	20	184	

Abbreviations: Pt, patient; Pv, previous; ICB, immune checkpoint blockade; ART, antiretroviral therapy; M: male; F, female; NSCLC, non-small cell lung cancer; HL: Hodgkin lymphoma, Nivo, nivolumab; Pembro, pembrolizumab; Cemi, cemiplimab; NA, Not Available, %, percentage. In the line named “all”, absolute values are medians.

## Data Availability

All biological data were linked and shared at Department of Immunology, Pitié-Salpétrière Hospital. External users with a formal analysis plan can request access to the data to Department of Immunology.

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
