# Peer review of "Impact of Anti PD-1 Immunotherapy on HIV Reservoir and Anti-Viral Immune Responses in People Living with HIV and Cancer"

_cells, 2022, doi:10.3390/cells11061015_

Round 1

Reviewer 1 Report

This study investigates the potential link between immune checkpoint blockade therapy and the size of HIV reservoir in PLWH. Samples were collected at different time points from HIV-1 positive individuals who also underwent ICB to treat various types of cancers. Some patients displayed reduced levels of cell-associated HIV DNA after cycle 2, so the authors grouped the subjects into RR (reservoir reduction) vs. No RR groups (those with no significant reduction) and compared the two for various immunological markers. It is worth mentioning that the majority of patients did not display any change in cell-associated DNA. Baseline HIV DNA was higher in the RR group, but there was no significant difference in the blood cell counts, cellular activation status, viral loads, HIV-specific T cells or anything that has to do with other viral infections. There is some evidence regarding compensatory mechanisms perhaps reducing the effect of ICP blockade due to the upregulation of other immune checkpoint molecules.

I have one major issue and that is the overall conclusion of this study is not clear to me. The authors mention that there have been conflicting reports regarding the effect of immune checkpoint blockade on HIV infection. It seems to me that the current study adds to that controversy, as there is no consistent or significant change that is caused by ICB in their cohort. The majority of the patients did not have altered reservoir size anyway, and the minority that did have not differentially regulated anything in the course of their HIV-1 infection. This tells me that ICB is really not a viable or effective option for HIV-1 cure strategies at all.

Each time one talks about “HIV infection,” it is best to refer to it as HIV-1, since this is the official name of the virus. I am guessing the authors are referring to HIV-1 and not HIV-2, as the latter is quite limited geographically. These should be corrected in the text. It would also be good to include some information about the groups/subtypes of HIV-1 infections observed in the patients. Are they all infected with the same subtype/clade?

Some minor points

Line 164 states: “median HIV VL was 20 cps/mL (range 20-352).” In the table there are some patients with less than 20 or undetectable (<1 copies/ml) viral loads. Do these not get included in the range?

Line 176, there is no red font. In the table dolutegravir is sometimes written with an accent.

Line 188, superscript needed for 10^6

Line 269; “stained with intracellular cytokine” ?? you mean IFN-γ-specific antibodies?

Line 377, 387; I believe the correct abbreviation for people living with HIV is PLWH, and not PLWHIV

Reviewer 2 Report

The authors evaluated the effects of immune checkpoint blockade with anti-PD1 antibodies on HIV reservoirs and HIV-specific immunity in cohort of 32 PLWH with cancer. Changes in cell associated viral DNA in the blood were used to measure anti-PD1-treatment-induced virus purging and clearance of infected cells by HIV-specific CTLs. No significant differences in immune activation and HIV-specific T cell responses were observed. Importantly, an increase of the expression of other immune checkpoint proteins such as CTLA-4 and TIM-3 was observed on virus specific CD8+ T cells indicating anti-PD1-treatment-induced ICP compensation. Additionally, authors demonstrated no adverse effects of anti-PD1-treatment on viral co-infections. The manuscript is well written, the data are clearly shown, and the introduction and discussion are complete and comprehensive. The topic is timely and interesting, demonstrating the failure of ICB monotherapy is important to inform future clinical trials and HIV cure therapies.

Author Response

We would like to thank the reviewer for his comments on our manuscript. There was no issue to address. 

Reviewer 3 Report

Authors studied the impact of anti-PD-1 therapy on HIV reservoirs and anti-viral immune responses in people living with HIV treated for cancer. This study found that the HIV-DNA tended to decrease early but CTLA-4+CD4+ T cells were increased in all patients and a greater increase observed in CTLA-4+ and TIM-3+CD8+ T cells of patients without HIV-DNA reduction compared to the others. This is encouraging as this study shows that the use of ICB for the cancer patients who are living with HIV, do not affect the immune response.

Main drawback of this study is avoiding the age factor into consideration while analyzing. The RR category identified in this study tends to fall in the lower age group (<60).  In fact, the study cohort seems to have higher baseline HIV DNA in the lower age group (<60yr old) with an average median of ~250 compared to the older group (>60yr) that has a median value ~132. Whether the observation that the greater increase in CTLA-4+ and TIM-3+CD8+ T cells in patients without HIV-DNA reduction compared to the others is also influenced by age should be carefully investigated.

Minor points:  

  1. In table 1 the patients with reservoir reduction are not highlighted though mentioned in the legend.
  2. Figure 7 title repeats the word ‘virological’
  3. Explain why several of the data points in Fig S1B is clustered in straight line (why many patients have the exact same RNA copies). Similarly, S14B, D, and E needs an explanation for the exact same values for viral loads.
  4. Information on how many technical replicates were performed for each of these experiments should be described under the figure legend.
